# Effect of Calcination Temperature on Mechanical Properties of Magnesium Oxychloride Cement

**DOI:** 10.3390/ma15020607

**Published:** 2022-01-14

**Authors:** Chenggong Chang, Lingyun An, Rui Lin, Jing Wen, Jinmei Dong, Weixin Zheng, Fengyun Yan, Xueying Xiao

**Affiliations:** 1State Key Laboratory of Advanced Processing and Recycling of Non-Ferrous Metals, Lanzhou University of Technology, Lanzhou 730050, China; ccg168@isl.ac.cn; 2Key Laboratory of Comprehensive and Highly Efficient Utilization of Salt Lake Resources, Qinghai Institute of Salt Lake, Chinese Academy of Sciences, Xining 810008, China; ccg168@163.com (R.L.); wj580420@isl.ac.cn (J.W.); dongda839@isl.ac.cn (J.D.); zhengweixin@isl.ac.cn (W.Z.); 3Key Laboratory of Salt Lake Resources Chemistry of Qinghai Province, Xining 810008, China; 4College of Physics and Electronic Information Engineering, Qinghai University for Nationalities, Xining 810007, China; anlingyun0825@126.com

**Keywords:** magnesium oxychloride cement, calcination temperature, microstructure, phase composition, mechanical properties

## Abstract

In order to make full use of magnesium chloride resources, the development and utilisation of magnesium oxychloride cement have become an ecological and economic goal. Thus far, however, investigations into the effects on these cements of high temperatures are lacking. Herein, magnesium oxychloride cement was calcinated at various temperatures and the effects of calcination temperature on microstructure, phase composition, flexural strength, and compressive strength were studied by scanning electron microscopy, X-ray diffraction, and compression testing. The mechanical properties varied strongly with calcination temperature. Before calcination, magnesium oxychloride cement has a needle-like micromorphology and includes Mg(OH)_2_ gel and a trace amount of gel water as well as 5 Mg(OH)_2_·MgCl_2_·8H_2_O, which together provide its mechanical properties (flexural strength, 18.4 MPa; compressive strength, and 113.3 MPa). After calcination at 100 °C, the gel water is volatilised and the flexural strength is decreased by 57.07% but there is no significant change in the compressive strength. Calcination at 400 °C results in the magnesium oxychloride cement becoming fibrous and mainly consisting of Mg(OH)_2_ gel, which helps to maintain its high compressive strength (65.7 MPa). When the calcination temperature is 450 °C, the microstructure becomes powdery, the cement is mainly composed of MgO, and the flexural and compressive strengths are completely lost.

## 1. Introduction

The Qaidam Basin in Qinghai Province, China is extremely rich in magnesium chloride resources; the existence of 4.82 billion tons of this mineral have been verified in this area. Initially, utilisation of this mineral resource was mainly focused on potash development. For every ton of potassium chloride, 8–10 t of magnesium chloride is produced as a by-product. The annual production of potassium chloride is approximately 5 × 10^6^ t, and hence the by-product of this is 4–5 × 10^7^ t of magnesium chloride. After years of potassium extraction, Qinghai has accumulated a large amount of bischofite, the main component of which is MgCl_2_·6H_2_O. These magnesium chlorides have not been fully exploited and hence they have accumulated for a long time. In addition, they can cause ‘magnesium damage’, which seriously affects the surrounding environment. As well as using magnesium chloride to produce a series of magnesium products—such as magnesium metal, anhydrous magnesium chloride, magnesium hydroxide, and magnesia whiskers—the development and utilisation of magnesium oxychloride cement have become an ecological, economic, and effective means of solving the problem of magnesium chloride accumulation [1,2].

Magnesium oxychloride cement (MOC), also known as magnesium cement, is a slurry prepared by combining a specific mass ratio of light burned magnesia powder and magnesium chloride solution. After solidification, it has superior mechanical properties [3,4], and it is a ternary system, MgO–MgCl_2_–H_2_O, at room temperature. The formation mechanism involves the active MgO being first dissolved with Mg^2+^ and OH^−^ ions in the MgCl_2_ aqueous solution before Mg^2+^, Cl^−^, OH^−^, and H_2_O are directly reacted in the slurry system. The reaction produces 5Mg(OH)_2_·MgCl_2_·8H_2_O (abbreviated as ‘the 5·1·8 phase’ or P5), 3Mg(OH)_2_·MgCl_2_·8H_2_O (abbreviated as ‘the 3·1·8 phase’ or P3), and Mg(OH)_2_ gel [5,6]. When the active MgO/MgCl_2_ molar ratio is greater than six, the resulting product is 5Mg(OH)_2_·MgCl_2_·8H_2_O and Mg(OH)_2_ gel. Among these compounds, 5Mg(OH)_2_·MgCl_2_·8H_2_O mostly consists of needle-shaped microstructures, whereas the morphology of Mg(OH)_2_ gel is mostly scaly and fibrous [7,8]. Studies have shown that the 5·1·8 phase is the main strength phase of magnesium oxychloride cement, and this phase provides the mechanical properties for magnesium oxychloride cement [9].

At the same time, magnesium oxychloride cement is a type of air-hardening cementitious material; because of its advantages of fast hardening, early strength, sound insulation, and simple preparation process, it is widely used in arts and crafts, composite panels, construction materials, magnesium cement flue, and so on, and the output value of magnesium oxychloride cement products in China exceeds RMB 50 billion [10]. In the future, with the adjustment of China’s energy sector, magnesium oxychloride cement-related products are expected to have broader application prospects and huge market potential [11,12]. However, once a fire occurs, the temperature of a room can reach more than 400 °C. Furthermore, in accordance with the standard of “Code for Fire Protection in Building Design” (GB50016-2006), magnesium cement flue should be non-combustible bodies with a fire resistance rating of not less than 1.00 h. Therefore, it is important to study the high-temperature performance of magnesium oxychloride cement. At present, most scholarly research has mainly focused on the fields of water resistance [13,14,15,16,17,18] and composite functional materials [19,20,21,22,23], and there is a lack of reports on the application of magnesium oxychloride cement under the high-temperature conditions or of the impact of high temperature on its properties.

Therefore, to study the failure mechanism of magnesium oxychloride cement at high temperatures, the study reported herein is concerned with the use of different temperatures to calcinate magnesium oxychloride cement samples. The relationships between the calcination temperature and macroscopic morphology, microstructure, mass loss, and mechanical properties of the sample are investigated. In addition, the change in the mechanical properties of magnesium oxychloride cement with temperature is analysed, and its failure mechanism is explored.

## 2. Experimental Section

### 2.1. Principal Raw Materials

The main raw materials used in this experiment were bischofite from Golmu, Qinghai Province, and light burned magnesia from Haicheng, Liaoning Province. Among them, the active ingredient of light burned magnesia is 50.51%. The chemical compositions of these two raw materials are listed in Table 1 and Table 2.

### 2.2. Preparation

Firstly, 23.5 wt.% MgCl_2_ aqueous solution was prepared. The activity MgO of light burned magnesia was 50.51%. According to the fact that the molar ratio of active MgO, MgCl_2_ and H_2_O was 7.7:1:17, 180 g of light burned magnesia and 120 g of magnesium chloride aqueous solution were mixed and then stirred for 3 min to form a slurry, which was injected into a mould with the size of 40 mm × 40 mm × 160 mm. After hardening under ambient laboratory conditions for 24 h, the samples were taken out and then cured for 28 days. The ambient laboratory was 25 ± 2 °C, and its humidity was 33 ± 5%.

### 2.3. Calcination Experiments

After 28 days of curing under ambient laboratory conditions (at approximately 25 °C), the magnesium oxychloride cement was placed in a box-type resistance furnace (KSL-1700X, HE FEI KE JING MATERIALS & TECHNOLOGY CO., LTD., Hefei, China) and heated at 5 °C/min to 100 °C, 200 °C, 300 °C, 400 °C, 450 °C, or 500 °C, respectively. Subsequently, it was maintained at this temperature for 2 h. After heating, the temperature was naturally cooled and then the magnesium oxychloride cement was removed from the furnace.

### 2.4. Characterisation

An electronic balance (SPX622ZH, OHAUS, Parsippany, NJ, USA) was used to test the mass of magnesium oxychloride cement that had been naturally cured for 28 days in the test room as well as calcined at various temperatures in the laboratory. A total of six samples were used at each temperature, and each sample was tested five times. Finally, its average value was calculated. The mean value under indoor natural curing was recorded as *m*_A_, and the mass value at each calcination temperature was recorded as *m*_B_. Hence, the mass loss of the magnesium oxychloride cement before and after calcination was *m*_A_ − *m*_B_, and the mass loss percentage was (*m*_A_ − *m*_B_)/*m*_A_; the bulk density after heating was *m*_B_/256 (g/cm^3^), in which 256 cm^3^ is the volume of the sample. Field-emission scanning electron microscopy (FE-SEM, SU8010, Hitachi High-Tech, Japan) equipped with Energy Dispersive Spectrometer (EDS, X-MAX, Oxford Instruments, Abingdon, UK) and X-ray diffraction (XRD, Model D8 Discover, Bruker, Bremen, Germany) was used to analyse the microstructure, element content and phase composition of the magnesium oxychloride cement samples. A total of six pieces of the randomly selected samples were used for testing the flexural and compressive strengths of the magnesium cement samples by using a micro-electro-hydraulic servo pressure testing machine (HYE-300B-D, Beijing sanyuweiye testing machine Co., Ltd., China) in accordance with the ‘Cement Mortar Strength Inspection Method (ISO Method)’ described in the Chinese national standard GB/T17671-2020. At the same time, their average values as well as the standard deviation were obtained.

## 3. Results and Discussion

### 3.1. Effect of Calcination Temperature on Morphology of Magnesium Oxychloride Cement

The photograph in Figure 1 illustrates the macroscopic morphology of the magnesium oxychloride cements prepared by natural curing at room temperature for 28 days and calcination at different temperatures. As shown in Figure 1a, the naturally cured magnesium oxychloride cement is milky white, the surface of the sample is complete and continuous, without any obvious cracks or defects. After calcination at different temperatures, the colour of the magnesium oxychloride cement changes significantly, and the surface colour appears to be different for each temperature (Figure 1b–g). At the same time, during the entire experiment, when the cement is calcinated at or above 200 °C, there is a pungent odour and hair-like cracks appeared in the samples. After calcination at 500 °C, wide fracture opening is apparent, as shown in Figure 1g, indicating that the magnesium oxychloride cement have completely failed at this calcination temperature.

### 3.2. Effect of Calcination Temperature on Micromorphology of Magnesium Oxychloride Cement

Figure 2 shows scanning electron microscopy (SEM) images that illustrate the micromorphology of the magnesium oxychloride cement samples cured for 28 days at room temperature and calcinated at different temperatures. It can be seen from Figure 2 that after 28 days of curing at room temperature, the magnesium oxychloride cement is made up of needle- or rod-shaped particles, which form an alternate network structure and can provide better mechanical properties. After calcination at 100 °C, the overall morphology remains the same. However, the rods become shorter and thicker. The reason for this change may be that at 100 °C, the bonding effect of the gel is lost. After calcination at 200 °C, some of the rod-like structures become shorter and even lost, and some fibrous and scaly morphology is apparent. After calcination at 300 °C, the rod-like structures have almost entirely disappeared and the fibrous morphology is staggered. After 400 °C calcination, the rod-like structures have disappeared completely, leaving only the scaly and fibrous morphology. After calcination at 450 °C, the morphology changes suddenly, the scaly and fibrous structures disappear, and fine particles appear. Finally, after calcination at 500 °C, the fine particles are still present. In summary, it can be concluded that calcination temperature has a significant effect on the micromorphology of magnesium oxychloride cement.

### 3.3. Effect of Calcination Temperature on Cl element Content in Magnesium Oxychloride Cement

Figure 3 shows the content of Cl element in the magnesium oxychloride cement at different calcination temperatures. It can be seen from Figure 3 that with the increase in the calcination temperature, the amount of Cl element gradually decreases. This is due to the decomposition of the 5·1·8 phase in the magnesium oxychloride cement, in which the Cl element is released in the form of HCl gas, resulting in a decrease in the content of Cl element in the magnesium oxychloride cement.

### 3.4. Effect of Calcination Temperature on Phase Composition of Magnesium Oxychloride Cement

Figure 4 shows the XRD patterns of magnesium oxychloride cements cured at room temperature for 28 days and calcinated at different temperatures. From the XRD pattern, it can be concluded that the magnesium oxychloride cement cured for 28 days at room temperature is mainly composed of the 5·1·8 phase and Mg(OH)_2_ (Brucite), and the diffraction peaks of MgCO_3_ (Magnesite), CaCO_3_ (Calcite), MgO (Periclase) and SiO_2_ (Quartz) originate from raw materials―light burned magnesia. After calcination at 100 °C, the phase composition of magnesium oxychloride cement does not change significantly. After calcination at 200 °C, the diffraction peak of the 5·1·8 phase is significantly weakened, and a faint diffraction peak assigns to the 3·1·8 phase, indicating that the proportion of the 5·1·8 phase is reduced and a new 3·1·8 phase is formed. After calcination at 300 °C, the 5·1·8 phase diffraction peaks become extremely weak, and the 3·1·8 phase diffraction peaks disappear. Instead, XRD peaks characteristic of MgCO_3_, Mg(OH)_2_, and MgO phases are apparent. After calcination at 400 °C, the XRD features assign to the 5·1·8 phase have disappeared completely. At this point, the magnesium oxychloride cement is mainly composed of Mg(OH)_2_, MgCO_3_, and MgO, and the Mg(OH)_2_ diffraction peaks increase, indicating that the Mg(OH)_2_ content increases. After calcination at 450 °C, the MgCO_3_ and MgO diffraction peaks increase slightly, and the Mg(OH)_2_ diffraction peaks decrease significantly, indicating that the Mg(OH)_2_ content of the sample has been significantly reduced by this stage. After calcination at 500 °C, the XRD pattern of the magnesium oxychloride cement is mainly composed of the MgCO_3_ and MgO diffraction peaks, with the Mg(OH)_2_ diffraction peaks also having disappeared completely.

### 3.5. Effect of Calcination Temperature on Mass Loss Percentage and Bulk Density of Magnesium Oxychloride Cement

Figure 5 shows the mass loss percentage and bulk density of magnesium oxychloride cement calcinated at different temperatures. It can be seen from Figure 5 that the mass loss percentage of magnesium oxychloride cement gradually increases with the calcination temperature, and the bulk density decreases accordingly. After calcination at 100 °C, the mass loss percentage of magnesium oxychloride cement is 0.52%, and its bulk density is 1.93 g/cm^3^, and when calcined at 500 °C, the mass loss percentage is 30.66%, and its bulk density is reduced by 29.74% compared to that of the uncalcinated sample. This shows that the calcination temperature has a significant effect on the mass loss percentage and bulk density of the sample, and the higher the calcination temperature, the lower the quality, the smaller the density, and the looser the magnesium oxychloride cement. These changes may result in the mechanical properties of the cement being worsened.

### 3.6. Effect of Calcination Temperature on Mechanical Properties of Magnesium Oxychloride Cement

It can be seen from Table 3 that the flexural and compressive strengths of magnesium oxychloride cement cured for 28 days at room temperature are 18.4 MPa and 113.3 MPa, respectively. After calcination at 100 °C, the compressive strength of magnesium oxychloride cement increases slightly to 119.7 MPa, but the flexural strength decreases by 57.07%. When calcined at 200 °C, the flexural and compressive strengths of the cements decrease significantly. After calcination at 300 °C, the flexural strength of the magnesium oxychloride cement is only 3.1 MPa, while the compressive strength is as high as 96.9 MPa. When 400 °C is used as the calcination temperature, the flexural strength of the resultant magnesium oxychloride cement almost completely disappears, and the compressive strength drops to 65.7 MPa, which is 57.99% of its compressive strength at room temperature. After calcination at 450 °C, the flexural strength disappears, and the compressive strength is only 7.9 MPa. The flexural and compressive strengths disappear completely when 500 °C is the calcination temperature. In this case, the magnesium oxychloride cement completely loses its mechanical properties and fails.

In summary, the calcination temperature has a significant effect on the mechanical properties of magnesium oxychloride cement. When cured at room temperature for 28 days, the magnesium oxychloride cement is milky white, it has excellent macroscopic surface integrity, and the microscopic surface is characterised by needle- or rod-like alternate network structures. It is mainly composed of the 5·1·8 phase and Mg(OH)_2_ gel, and its bulk density is as high as 1.95 g/cm^3^. This condition corresponds to the highest flexural and compressive strengths among the samples analysed in this study.

When calcine at 100 °C, the micromorphology of the magnesium oxychloride cement changes, the slender needle- or bar-like structures are converted into stubby rod shapes, and the mass loss percentage is 0.52%, which corresponds to the loss of the gel water stored in the component of the material. The gel water exists in the pores of the 5·1·8 phase gel and is adsorbed on the surface of the colloidal particles; it is tightly bound to the gel crystals. It acts as a connection between the 5·1·8 phase gel and between the 5·1·8 phase gel and other phases, and so its presence is helpful to improve flexural strength for the magnesium oxychloride cement. Therefore, after the magnesium oxychloride cement is calcined at 100 °C, the gel water detaches from the surface of the gel and breaks the connection between the gels, causing the magnesium oxychloride cement micromorphology to be altered, with the stubby rod structures replacing slender pin-rod shaped structures. As a result, the flexural strength of the magnesium oxychloride cement decreases significantly, by 57.07% with respect to that at room temperature.

When calcined at 200 °C, a scaly and fibrous morphology appears in magnesium oxychloride cement, and the proportion of the 5·1·8 phase is reduced. At the same time, the mass loss percentage of the sample is 4.74%. Beside the loss of 0.5% of the gel water, the residual loss of 4.22% may be ascribed to the fact that a small part of the 5·1·8 phase decomposes into HCl gas; this is consistent with the phenomenon of irritant gas volatilisation during the heating process.

The decomposition formula is as follows:(1)$$5\;\mathrm{Mg}(\mathrm{OH}{)}_{2}\cdot {\mathrm{MgCl}}_{2}\cdot 8H_{2}O\;\overset{\mathrm{heat}}{\to }\;3\mathrm{Mg}(\mathrm{OH}{)}_{2}\cdot {\mathrm{MgCl}}_{2}\cdot 8H_{2}O+\mathrm{Mg}(\mathrm{OH}{)}_{2}+\mathrm{HCl}\;(↑)$$

The 5·1·8 phase is the main strength phase of magnesium oxychloride cement, and this can provide excellent mechanical properties for the cement. Therefore, the reduction in the 5·1·8 phase content results in a significant decrease in the flexural and compressive strengths of magnesium oxychloride cements.

After calcination at 300 °C, the needle-like morphology of magnesium oxychloride cement disappears, and areas of scaly and fibrous structures are formed. The 5·1·8 phase content is extremely low, and, at this time, the main phases existing in the sample are MgCO_3_ (Magnesite), Mg(OH)_2_ (Brucite), and MgO (Periclase). The flexural strength is very small, but the compressive strength remains as high as 96.9 MPa. This is because, in addition to the 5·1·8 phase, there are other phases that also provide compressive strength, and they together provide the gelling properties of the magnesium oxychloride cement, while MgO and MgCO_3_ play a role as aggregates such as sand and stone. Finally, similar to the concrete, the magnesium oxychloride cement possesses a higher compressive strength but lower flexural strength.

After calcination at 400 °C, the micromorphology of the magnesium oxychloride cement is entirely scaly and fibrous, and proportion of fibrous structure is obviously increased with respect to the lower-temperature calcination products. The XRD spectrum confirms that the 5·1·8 phase in the magnesium oxychloride cement has completely disappeared. Instead, the sample is mainly composed of Mg(OH)_2_, MgCO_3_, and MgO phases, and the Mg(OH)_2_ content is significantly increased. At this calcination temperature, while the flexural strength is almost completely lost, the compressive strength drops to 65.7 MPa; this is still 57.99% of the value measured for the room-temperature sample. This result further confirms that in addition to the 5·1·8 phase, which provides part of the compressive strength, some other phases also provide the cement with compressive strength.

After calcination at 450 °C, the microscopic morphology of magnesium oxychloride cement changes suddenly, and the fibrous morphology disappears. Although the presence of phases such as MgCO_3_ and MgO are apparent, the Mg(OH)_2_ diffraction peaks are significantly reduced for this sample, indicating that the content of Mg(OH)_2_ is relatively low. The compressive strength decreases to 7.9 MPa, suggesting that the formation and relative amount of Mg(OH)_2_ are important influences on the compressive strength.

After calcination at 500 °C, the fibrous microstructures in magnesium oxychloride cement are completely lost, and the Mg(OH)_2_ diffraction peaks also disappear. From this, it can be implied that the fibrous structures in magnesium oxychloride cement may be Mg(OH)_2_. In addition, as the disappearance of the Mg(OH)_2_ diffraction peak coincides with the loss of compressive strength, it is suggested that the compressive strength is strongly dependent on the Mg(OH)_2_ content.

To further confirm the relationship between the Mg(OH)_2_ content and compressive strength of magnesium oxychloride cement, a semi-quantitative analysis of the XRD spectra (Figure 4) by using the Topas 4.2 software (Bruker, Germany) is used to determine the content of 5·1·8 phase and Mg(OH)_2_ phase, which are compared with the compressive strength, as shown in Figure 6. It can be seen from Figure 6 that the content of the 5·1·8 phase decreases as the calcination temperature increases, and 5·1·8 phase almost disappears after calcination at 300 °C, but the compressive strength at this stage is as high as 96.9 MPa. After calcination at 400 °C, the 5·1·8 phase does not exist, but the Mg(OH)_2_ content is the highest among the samples. For this sample, the compressive strength is still 65.7 MPa. When, at 500 °C, the Mg(OH)_2_ phase disappears, the compressive strength also drops to zero. These clearly demonstrate that Mg(OH)_2_ can provide compressive strength for magnesium oxychloride cement. Therefore, based on the relationship between the calcination temperature and mechanical properties of magnesium oxychloride cement, it is further confirmed that the 5·1·8 phase in magnesium oxychloride cement provides some of the flexural and compressive strengths of the material at room temperature. The small amount of gel water in magnesium oxychloride cement plays a role in connecting the gels, which is beneficial for improving the flexural strength, but it has little effect on the compressive strength. The presence of a small amount of Mg(OH)_2_ gel also provides some of the compressive strength for magnesium oxychloride cement, but it cannot provide flexural strength. Therefore, the 5·1·8 phase, gel water, and Mg(OH)_2_ gel together produce the mechanical properties of magnesium oxychloride cement.

## 4. Conclusions

To make full use of magnesium chloride in Qinghai Salt Lake, expand the application fields of magnesium oxychloride cement, and explore the failure mechanism of magnesium oxychloride cement at high temperatures, the different temperatures were used to calcinate the pure slurry samples of magnesium oxychloride cement. The effects of calcination temperature on microstructure, phase composition, flexural strength, and compressive strength of magnesium oxychloride cement were studied in this study. The main conclusions are as follows:As the calcination temperature increased, the macroscopic morphology of the magnesium oxychloride cement changes significantly. At the same time, the microscopic morphology changes from being principally needle- or bar-like to fibrous and then powdery, and the phase composition changes from the 5·1·8 phase being dominant to the Mg(OH)_2_ gel dominating and then to MgO being dominant. The mechanical properties change significantly with these structural and compositional changes.At room temperature, magnesium oxychloride cement mainly consists of fine needle-like structures of the 5·1·8 phase with some Mg(OH)_2_ gel and a trace amount of gel water. The flexural strength is 18.4 MPa and the compressive strength is 113.3 MPa. After calcination at 100 °C, owing to the volatilisation of the gel water, the mass loss percentage is 0.52%, which causes flexural strength to decrease by 57.07%. When calcination is carried out at 400 °C, the magnesium oxychloride cement becomes fibrous. The main chemical component is Mg(OH)_2_ gel, which can improve the compressive strength of the materials, and hence the compressive strength remains high at 65.7 MPa. When the calcination temperature is 450 °C, a powdery microstructure is observed in the product, which is mainly composed of MgO. In this case, the flexural and compressive strengths are almost entirely lost, and the magnesium oxychloride cement fails.

## Figures and Tables

**Figure 1 materials-15-00607-f001:**
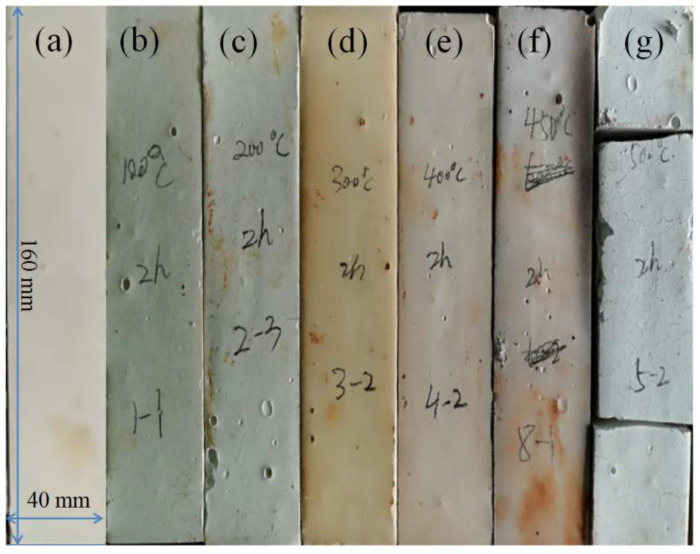
Macromorphology of MOC (**a**) cured for 28 days at room temperature and then heated at (**b**) 100 °C, (**c**) 200 °C, (**d**) 300 °C, (**e**) 400 °C, (**f**) 450 °C, and (**g**) 500 °C.

**Figure 2 materials-15-00607-f002:**
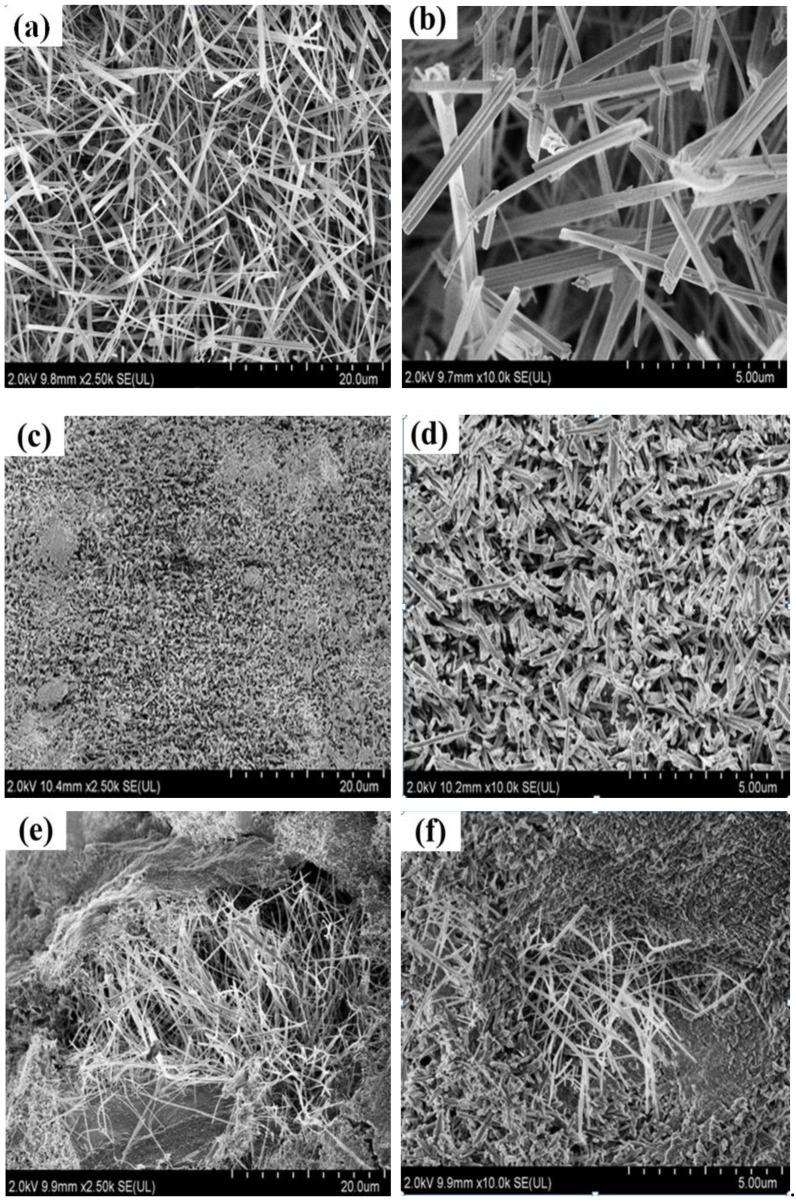
SEM images showing micromorphology of MOC (**a**,**b**) cured for 28 days at room temperature and then heated at (**c**,**d**) 100 °C, (**e**,**f**) 200 °C, (**g**,**h**) 300 °C, (**i**,**j**) 400 °C, (**k**,**l**) 450 °C, and (**m**,**n**) 500 °C.

**Figure 3 materials-15-00607-f003:**
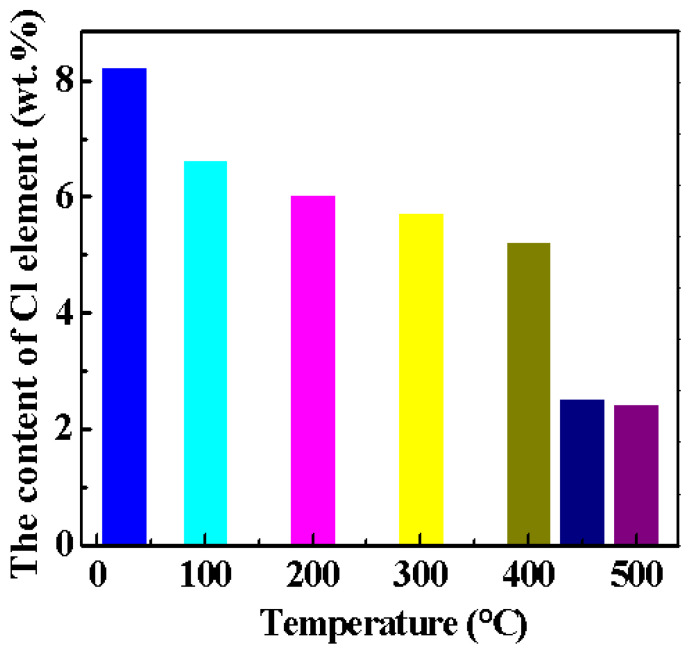
Content of Cl element in MOC after being heated at various temperatures.

**Figure 4 materials-15-00607-f004:**
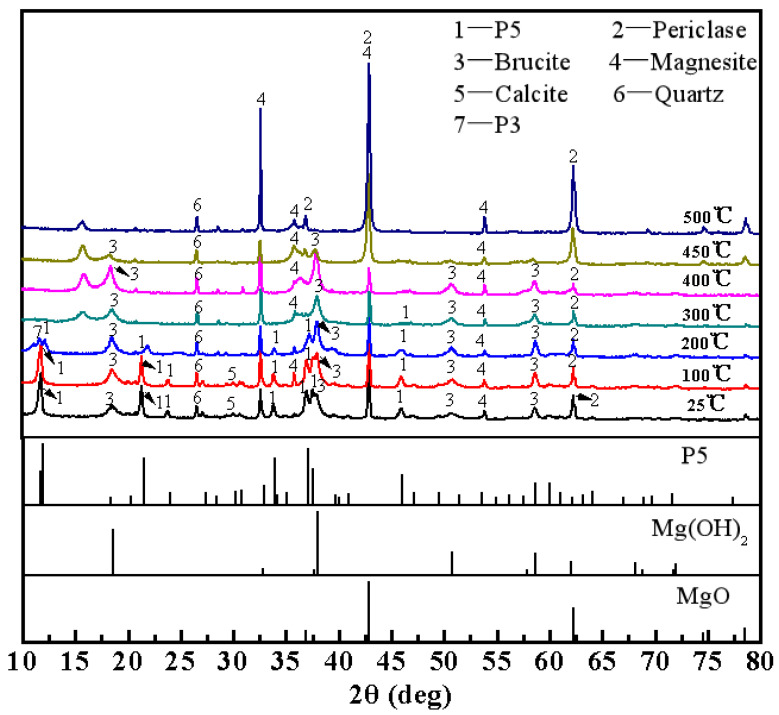
XRD patterns of MOC cured for 28 days at room temperature and calcinated at different temperatures.

**Figure 5 materials-15-00607-f005:**
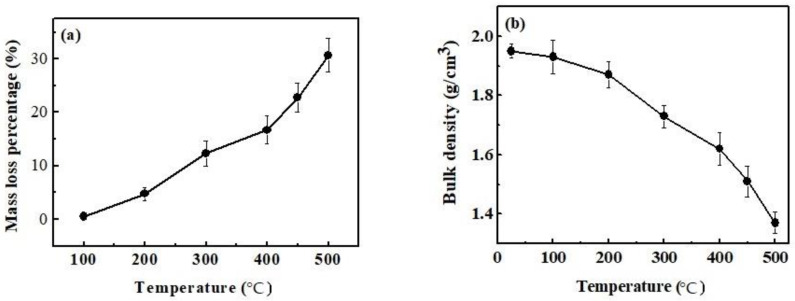
(**a**) Mass loss percentage and (**b**) bulk density of MOC calcinated at different temperatures.

**Figure 6 materials-15-00607-f006:**
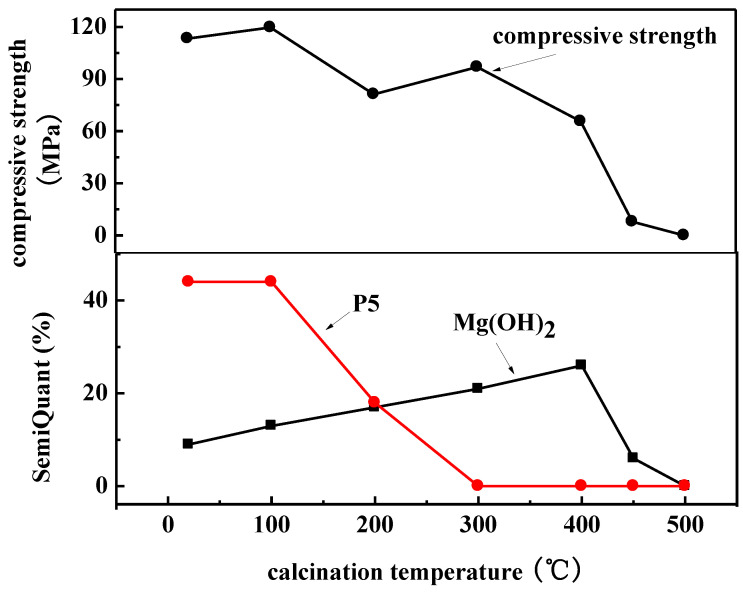
Relationship between compressive strength and P5 and Mg(OH)_2_ contents of MOC prepared using different calcination temperatures; the semiquantitive P5 and Mg(OH)_2_ contents (lower panel) are obtained from an analysis of the XRD spectra by using Topas 4.2 software.

**Table 1 materials-15-00607-t001:** Chemical composition of bischofite.

Composition	MgCl_2_	NaCl	MgSO_4_	KCl	CaCl_2_	Water Insoluble Matter	H_2_O
Content (wt.%)	44.90	0.13	0.06	0.01	0.03	0.27	54.6

**Table 2 materials-15-00607-t002:** Chemical composition of light burned magnesia.

Composition	MgO	MgCO_3_	CaCO_3_	f-CaO	Acid Insoluble Matter
Content (wt.%)	69.52	19.80	1.34	0.38	8.41

**Table 3 materials-15-00607-t003:** Effects of calcination temperature on mechanical properties of magnesium oxychloride cement.

Calcination Temperature/°C	Time/h	Flexural Strength/MPa	Standard Deviation (σ)	Compressive Strength/MPa	Standard Deviation (σ)
25	–	18.4	1.6892	113.3	1.9563
100	2	7.9	2.3160	119.7	2.0142
200	2	10.9	2.4562	81.2	2.3624
300	2	3.1	3.1156	96.9	2.8621
400	2	–	–	65.7	3.5610
450	2	–	–	7.9	3.9852
500	2	–	–	–	–

## Data Availability

The data presented in this study are available on request from the corresponding author.

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
