# Peer review of "Effect of Calcination Temperature on Mechanical Properties of Magnesium Oxychloride Cement"

_materials, 2022, doi:10.3390/ma15020607_

Round 1

Reviewer 1 Report

The authors have carried out a large amount of research and a significant number of results have been obtained, but some points require clarification:

  1. There is no justification for the need for research, except that it has not been previously investigated. In what conditions will the products be used and will there be the likelihood of such temperatures occurring?
  2. Table 1 shows the chemical analysis of raw materials; the amount of oxides is 45.4%. It is necessary to clarify what constitutes the remaining 54.6%.
  3. It is necessary to clarify the reasons for the choice of the temperature at which the study was carried out and the upper limit of 500 oС. Why research has not been carried out at higher temperatures, since the temperature in a building during a fire can reach higher values.
  4. L94 «… 25°Be magnesium chloride…» the symbol may not be displayed correctly. Please clarify or make corrections
  5. Line 95: it is necessary to indicate what binder/water ratio the solution was prepared
  6. Line 96: The text indicates that the shape of the samples is a cube. However, the given dimensions indicate that the shape of the samples is prismatic. You should check this information in the text
  7. Line 97: It is necessary to indicate at what humidity the experimental samples were stored before testing.
  8. In the Materials and Methods section, it is indicated that 6 samples were used for each temperature. You must specify the standard deviation for the obtained compressive and flexural strengths.
  9. Figure 3. It is recommended to increase the size of the figure, provide captions for all peaks (see attached file)
  10. Figure 4. Incorrect display of data (see attached file)
  11. The authors in the study do not consider the issue of volumetric deformations of the material when exposed to elevated temperatures, while a large weight loss due to dehydration of crystalline compounds is usually accompanied by shrinkage. The shrinkage phenomena of the cementitious matrix in composite materials, as a rule, proceed unevenly and lead to warpage or cracking of the product. It would be desirable to consider this issue in the article (at the discretion of the authors).
  12. Also, it is of great practical interest to study the issue of changes in material properties during subsequent moistening after exposure to high temperatures: will the restoration of properties occur and to what extent; boundary temperature at which the material can be operated further; whether there will be any destructive phenomena during humidification (for example, swelling). It is recommended to add this information to the article (at the discretion of the authors).
  13. The lack of information specified in paragraphs 11 and 12 makes it difficult to understand the practical use of the results obtained. It is recommended to clearly formulate the areas of practical use of the results obtained.

Reviewer 2 Report

Title: Effect of calcination temperature on mechanical properties of magnesium oxychloride cement

Journal: Materials

Manuscript ID: materials-1474186 - Review Request Reminder

This paper covers the effect of calcination temperature on mechanical properties of magnesium oxychloride cement.

The presented work has good scientific soundness. Nevertheless, I have to rise some questions / modifications to must be done.

SEM section. The Authors should add the EDS analysis.

XRD. Please identify all the peaks presented. The reference cards should be provided.

Fig. 4; unit to X axis must be corrected.

Decomposition of magnesium oxychloride should be commented based on the DTA-TG results at least.

The standard deviation must be added to mechanical strength results.

Why there is no dependence in the strength of the samples, e.g. 25, 100 and 200°C? How the Authors can explain an increase in mechanical strength in samples calcined between 100 and 200°C?

There is no clear correlation between the mechanical strength and other results.

In conclusion I can not recommend this work in the present form.

Reviewer 3 Report

This work is really very interesting and novel. The efforts of the researchers are appreciated however; some clarifications are required as follows:

Line 65-70 needs citation.

Line 98: Do not start a sentence with a digit.

Hydraulic cementitious composites are cured for 28 days in water. Why magnesium oxychloride cements in this study are cured for 28 days in air? Any reason… Do they gain complete strength in 28 days? If yes, how and provide any reference?

Line 103-104: Kindly elaborate the mode of curing i.e. in open air under controlled laboratory conditions.

Line 182-183: The calcination emits HCl gas. Does it not make the use of magnesium oxychloride cement a huge environmental hazard in case of fire?

Line 231: Why there is increase in compressive strength at 100ºC, although flexural strength decreases?

Table 3: There is huge decrease in compressive strength from 400-450ºC?

Line 258-259: There is improvement in compressive strength at 100C and not in flexural strength as demonstrated in Table 3 as well as in Lines 231-233. Kindly correct.

Equation 1: Kindly write “heat” upon the arrow. Is there any reference for this chemical reaction or this is just, you deduced from XRD?

Line 283: Kindly mention in brackets, the phases which provide gel properties.

Complete loss of flexural strength at 65 MPa compressive strength needs further elaboration. The complete loss of flexural strength at 7.9 MPa compressive strength is understandable.

Author Response

Please see the attchment. 

Reviewer 4 Report

Well-written paper, but I suggest some changes.

Line 14: the reason to use MOC is not "to prevent MgCl2 accumulation".

Line 100: from where comes this ratio of components?

Line 109: roasted is not proper word, better is heating, calcination or thermal load.

Line 177: how the Cl content was determined?

Line 191: SiO2, not SO2. But - any of your component is not containing SiO2. Check the raw materials and XRD.

All XRD results: XRD is providing information about phase composition, thus the names of phases must be used when the data are presented. For example, do not use CaCO3, but use calcite. Because, CaCO3 can be calcite, vaterite or aragonite.

Figure 5a: present just percentage, the mass loss in grams is not useful for readers.

Round 2

Reviewer 1 Report

All comments are taken into account. The article can be recommended for publication. 

Reviewer 2 Report

Fig. 8 - unit is still not corrected.
There is still some XRD peaks that have not been identified.
There is CaCO3 missing (5). The XRD analysis require deep reconsideration.
The standard deviations must be introduced into Figs.
I recommend reconsideration my last comments.
